# An unusual type of polymorphism in a liquid crystal

Lin Li[1], Mirosław Salamończyk [2], Sasan Shadpour[3], Chenhui Zhu[2], Antal Jákli[3] & Torsten Hegmann [1,3]

Polymorphism is a remarkable concept in chemistry, materials science, computer science, and biology. Whether it is the ability of a material to exist in two or more crystal structures, a single interface connecting to two different entities, or alternative phenotypes of an organism, polymorphism determines function and properties. In materials science, polymorphism can be found in an impressively wide range of materials, including crystalline materials, minerals, metals, alloys, and polymers. Here we report on polymorphism in a liquid crystal. A bent-core liquid crystal with a single chiral side chain forms two structurally and morphologically significantly different liquid crystal phases solely depending on the cooling rate from the isotropic liquid state. On slow cooling, the thermodynamically more stable oblique columnar phase forms, and on rapid cooling, a not heretofore reported helical microfilament phase. Since structure determines function and properties, the structural color for these phases also differs.

[1] Department of Chemistry and Biochemistry, Kent State University, Kent, OH 44242-0001, USA. [2] Advanced Light Source, Lawrence Berkeley National Laboratory, Berkeley, CA 94720, USA. [3] Chemical Physics Interdisciplinary Program, Liquid Crystal Institute, Kent State University, Kent, OH 44242-0001, USA. Correspondence and requests for materials should be addressed to A.Ják. (email: ajakli@kent.edu) or to T.H. (email: thegmann@kent.edu)

The ability of a solid material to exist in more than one form of crystal structures is called polymorphism. It can be found in many crystalline materials and has a broad relevance in fields such as pharmaceuticals and agriculture.

In terms of thermodynamics, there are two types of polymorphic behavior. For a "monotropic" system, the plots of the temperature-dependent free energies of the various polymorphs do not cross below all polymorphs' melts, i.e., any transition from one polymorph to another below melting will be irreversible. For an "enantiotropic" system, the plots of the temperature-dependent free energies of the various polymorphs show a crossing point before the various melting points. Two polymorphs can be interchanged reversibly by thermal pathways or through physical contact with a lower energy polymorph. Thermal annealing or slow cooling usually lead to the formation of more stable polymorph, whereas quenching (rapid cooling) may stabilize the metastable polymorph (Fig. 1). Apart from this, quenching is also used to freeze in (vitrify) the high temperature structure at low temperatures. The best-known examples for those are metallic glasses[1, 2] and high entropy alloys[3] formed at $10^6$ to $10^{12}$ °C s$^{-1}$ cooling rates[4]. Phase selection using controlled cooling rates in these materials is crucial to arrest (select) structures that translate into desirable macroscopic properties for various engineering applications. The vitrified amorphous structure, however, is not a polymorph; real polymorphs must have symmetries different from that of the melt they formed from on cooling.

In liquid crystals, 'rapid' cooling at rates on the order of 50–100 °C s$^{-1}$ achieved by simply removing the sample from a hotplate is frequently used to arrest a particular liquid crystalline phase as a glass circumventing crystallization[5]. Such vitrified structures are often utilized in freeze fracture transmission microscopy[6].

Examples for real polymorphism are very scarce in rod-shaped liquid crystals. One prominent example is the formation of an electric field-induced chiral smectic-C ferrielectric liquid crystal ($SmC^*_{FII}$) phase[7]. Recently Alaasar et al.[8] reported the formation of a Dark Conglomerate (DC) phase when a bent-core liquid crystal material was cooled slowly (<2 °C min$^{-1}$), and a crystalline phase when cooled at rates higher than 5 °C min$^{-1}$ from the isotropic liquid phase. In this case, therefore only one of the polymorphs is liquid crystalline, the other is crystalline. In bent-core liquid crystals it is often found that, depending on the cooling rate, one of the smectic liquid crystal phases is chiral ($SmC_aP_A$), the other one is racemic ($SmC_sP_A$)[9, 10]. However, apart from chirality and clinicity (anticlinic or synclinic) these two states have the same smectic structures and X-ray diffraction patterns.

In this paper, we present the first example of a liquid crystalline material forming entirely different phase structures, solely depending on the rate of cooling from the isotropic liquid phase.

Along with the ability to select the phase structure, the choice of cooling rate also facilitates a different structural color, which could be exploited in thermal sensor as well as other optical device and security applications.

## Results

**Materials.** Figure 2 shows the chemical structure and phase transition temperatures of compound **I**—a bent-core liquid crystal (LC) based on a *tris*-biphenyl diester core with one straight 1-octyloxy side chain on the *para*-side and one chiral (R)-2-octyloxy side chain on the *meta*-side of the central biphenyl ring. The two structurally related compounds **II** and **III** shown underneath are the parent compounds with either two achiral 1-octyloxy or two chiral (R)-2-octyloxy side chains, respectively. Their corresponding phase sequences and phase transition temperatures are also shown. The achiral compound **III** forms two enantiotropic LC phases (both on heating and cooling), a rectangular columnar phase (Col$_r$ or B1) at elevated temperatures and a modulated helical nanofilament phase (HNF$_{mod}$, from the category of B4 phases) at lower temperatures[11]. On further cooling, the HNF$_{mod}$ phase eventually forms a glassy state that persists well below room temperatures without subsequent crystallization. The homochiral compound **II** with two identical chiral side-chains only forms a dual modulated HNF phase (HNF$_{mod2}$) with an additional interlayer electron density modulation[11] (see Supplementary Information for description of the Helical Nanofilament Phase (HNF)[12] and a comparison of HNF[12], HNF$_{mod}$[13] and HNF$_{mod2}$ phases), which also does not crystallize upon cooling to temperatures well below the melting point measured on heating and rather forming a glass. On the basis of the data obtained from polarized optical microscopy (POM), differential scanning calorimetry (DSC), scanning electron microscopy (SEM), circular dichroism (CD) spectropolarimetry, atomic force microscopy (AFM), and small angle X-ray scattering

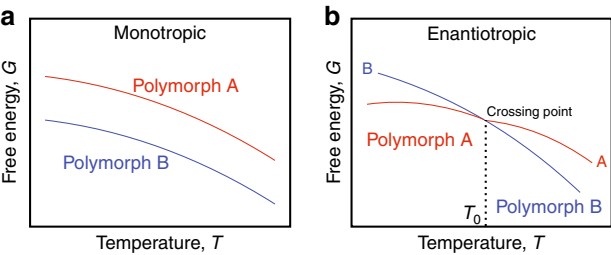

**Fig. 1** Schematic representation of the two types of polymorphs. **a** Monotropic and **b** enantiotropic polymorphs, where $T_O$ denotes the transition temperature in enantiotropic polymorphs (e.g., sulfur), where below the melting point one polymorph can reversibly transition to the other polymorph. For monotropic polymorphs (e.g., glyceryl stearates), this transition is not possible

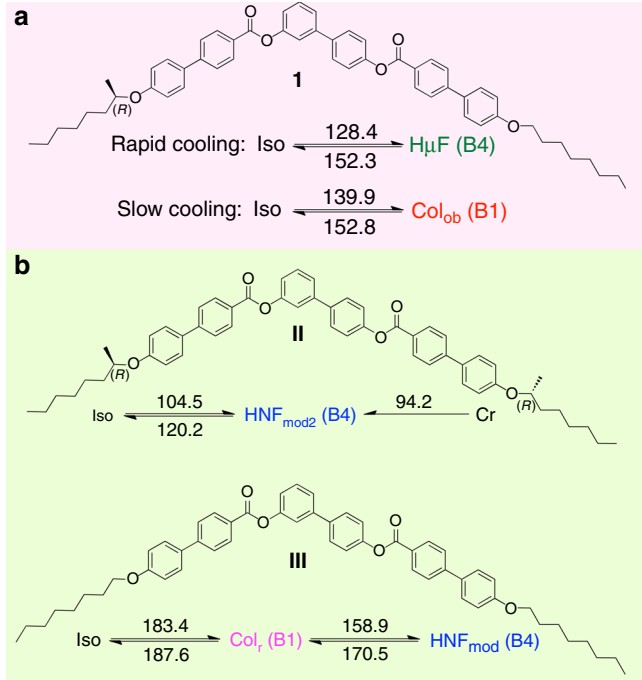

**Fig. 2** Materials and phase transition temperatures. Chemical structure of **a** compound **1** and **b** the two parent bent-core liquid crystal compounds **II** and **III**[11] with their respective phase sequences and transition temperatures in °C (determined by DSC measurements, peak values from 2$^{nd}$ heating and cooling run, respectively)

(SAXS), we will show that compound **1** with only one chiral side chain forms an oblique columnar (Col$_{ob}$) phase exclusively upon slow cooling (with cooling rates ranging from about 0.5 to 10 °C min$^{-1}$) and a new helical micro-filament phase (HμF) exclusively upon rapid cooling.

The syntheses of **II** and **III** were described earlier[11]. The synthesis of **1** involves the stepwise deprotection and esterification of mono-tetrahydropyranyl (THP)—mono-TBDMS (*tert*-butyldimethylsilyl)-protected 3,4'-dihydroxybiphenyl, which was synthesized by a Suzuki coupling[14, 15] reaction (Supplementary Notes 1–4 and Supplementary Figures 1–10).

**Polarized light optical microscopy.** First indication of a cooling rate-dependent formation of entirely different LC phase structures was provided by POM (Fig. 3) both under crossed and decrossed polarizers. Compound **1**, sandwiched between two pre-cleaned glass substrates, was first heated to the isotropic

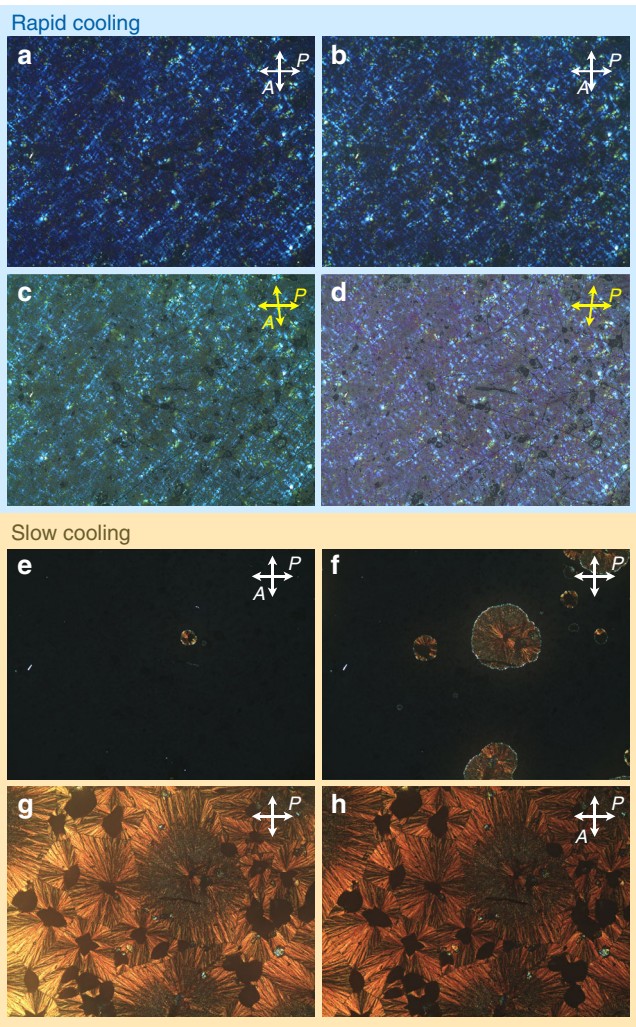

**Fig. 3** Polarized optical microscopy. Polarized optical photomicrographs of compound **1** after heating to the isotropic liquid phase (>153 °C). On heating, no LC phase or phase transition is detected: **a–d** upon rapid cooling (rate >50 °C min$^{-1}$) from the isotropic liquid phase the textures between crossed polarizers are shown in: **a** at 130 °C and **b** at 30 °C. **c**, **d** show the textures at 110 °C under decrossed polarizers. **e–h** Textures seen upon slow cooling at 2 °C min$^{-1}$: **e** at 146 °C, **f** at 144 °C, **g** at 142 °C, and **h** at 30 °C. See Supplementary Information (Supplementary Note 5, Supplementary Figure 13) for higher magnification/resolution images

liquid phase (>153 °C) and then immediately removed from the hotplate. Subsequent temperature-dependent POM imaging reveals grainy blue textures (both in transmission between crossed polarizers and reflection) similar to other HNF phase textures (Fig. 3a, b). Decrossing the polarizers by several degrees (Fig. 3c, d) did not indicate a conglomerate of left and right-handed homochiral domains, instead results in a uniform change of brightness for the entire homochiral thin film domain depending on the analyzer position (either rotated to the left or to the right). This is consistent with the existence of a chiral side-chain in the molecule.

In addition, the birefringence color of this rapidly quenched sample of **1** shows an astonishing temperature dependence (Supplementary Note 5, Supplementary Figure 11) even for temperature changes as small as a few °C. Such a large color change has not been previously reported for other HNF phase materials. Heating the sample again to the isotropic liquid state and subsequently cooling it slowly (at rates of 0.5, 2, or 5 °C min$^{-1}$) produces entirely different thin film textures (Fig. 3g, h). Now, feathered spherulitic or fan-shaped domains, frequently reported for B1-type phases of bent-core LCs[16], are formed. These do not show any change of brightness upon decrossing the polarizers (Supplementary Figure 12). Each phase obtained depending on the cooling rate is stable for several days, even weeks, without any indication of glass formation or crystallization.

**Differential scanning calorimetry.** DSC measurements were performed of material **1** in three different conditions: recrystallized from ethanol (neat), after heating the sample to the isotropic liquid phase followed by quenching, and after heating to the isotropic liquid phase and slow cooling at a rate of 5 °C min$^{-1}$. The resulting thermograms obtained from the first two processes are nearly identical in the first and second heating/cooling runs and only differ in the about 3 °C lower phase transitions recorded on cooling for the quenched sample (Supplementary Note 6, Supplementary Table 1 and Supplementary Figure 14a and Supplementary Figure 14b). It is indeed possible that some molecular associations from the microfilaments persist well into the isotropic liquid phase and that this causes this slight discrepancy in the phase transition temperatures between the cooling runs. The sample of **1** treated according to the last process, however, shows broader phase transitions with the phase transition on cooling now recorded about 11–14 °C lower than for the neat sample treated according to the first process. This last process used a heating/cooling rate of 50 °C min$^{-1}$ during the DSC experiment and a coexistence of the two phases observed on slow and rapid cooling, respectively, is expected (Supplementary Table 1 and Supplementary Figure 14c). All plots show just one phase transition peak on heating and cooling, indicative of a monotropic liquid crystal phase (only formed on cooling), with virtually equal phase transition enthalpies, but lower phase transition temperatures for each of the thermally-treated samples. The DSC data suggest that the polymorphs are energetically very close, including the crystalline phase formed by the neat sample of **1**.

**Circular dichroism spectropolarimetry and UV-vis spectroscopy.** The next set of experimental data were obtained by thin film CD spectropolarimetry (Fig. 4)[17]. CD experiments were previously used to determine the formation of HNF conglomerates[18] as well as homochiral HNF textures and phases, whereby a strong CD band at around 360 nm (positive or negative, depending on the sign of homochirality) indicated the chirality of the twisted layers stacked as HNFs, forming a secondary twist that depends on the handedness of the individual HNFs[11]. CD

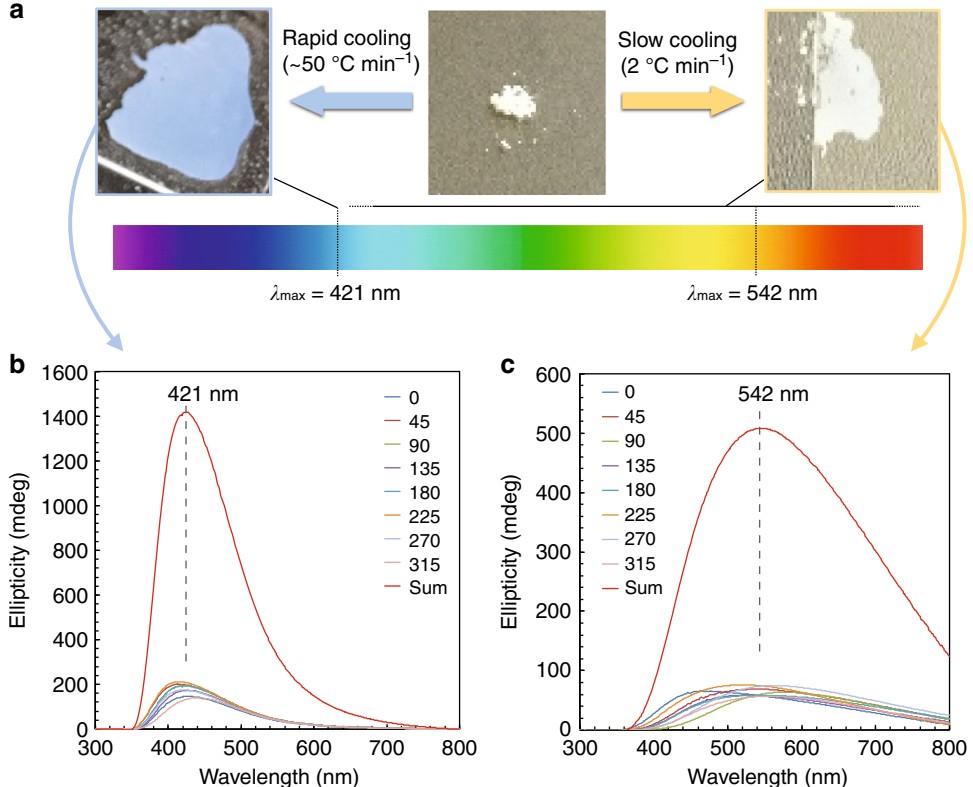

**Fig. 4** Optical characterization. **a** Photographs of samples of neat **1** after recrystallization, after rapid cooling showing blue structural color, and after slow cooling at a rate of 2 °C min⁻¹ showing a very faint yellowish color that is difficult to see. **b**, **c** Thin film CD spectra of 10 μm LC films of compound **1** between untreated quartz substrates on cooling at different cooling rates from the isotropic liquid phase at different sample rotation angles as indicated in the legends (the red curve is the sum of all spectra to cancel out linear dichroism and birefringence): **b** rapidly cooled (quenched) sample; **c** sample cooled at a rate of 2 °C min⁻¹. For larger plots of the individual CD spectra see Supplementary Note 7, Supplementary Figure 15

spectra of **1** at various sample rotation angles and the sum of all individual spectra are shown in Fig. 4b, c for the rapidly and slowly cooled samples, respectively. Note that the sum of the collected spectra cancels contribution from linear dichroism and birefringence Supplementary Note 7, Supplementary Figure 15. As suggested by POM, the sum CD spectrum of the rapidly cooled sample shows a band with a fairly sharp maximum at 421 nm indicating an HNF-type phase. In the case of the slowly cooled sample (Fig. 4c), the CD bands show maxima at longer wavelengths at the various sample rotation angles, and the sum spectrum shows a maximum at 542 nm featuring a much broader Full Width at Half Maximum (FWHM). The band of the sum spectrum practically covers the entire visible spectrum (Fig. 4c), which correlates to the visually observed faint yellow color of **1** after slow cooling from the isotropic liquid phase. At the same film thickness of ~10 μm, the maximum at 421 nm of the quenched sample is about 3-times larger than of the peak at 542 nm of the slowly cooled sample, however, the areas of the sum peaks are nearly equal. The exclusively positive CD signals indicate that both LC phases formed upon slow as well as rapid cooling from the isotropic liquid phase are chiral (or at least chiral features with chiral domains).

Transmission and reflection UV-Vis spectra (Supplementary Note 8, Supplementary Figure 16) confirm that the blue color observed for the quenched sample of **1** is a true reflection (i.e. structural) color indicated by a reflection peak at 430 nm in the reflection spectrum (Supplementary Figure 16a) and a broad transmission with a gradually increasing intensity at longer and longer wavelengths in the transmission spectrum (Supplementary

Figure 16b). As in the thin film CD spectra, the slow-cooled sample lacks these wavelength specific features.

**Small angle X-ray scattering**. To characterize the structures of compound **1** obtained both on rapid and slow cooling, we performed synchrotron SAXS experiments (see Methods section for more detail). A summary of the obtained SAXS data are shown in Fig. 5. On slow cooling from the isotropic liquid phase, **1** shows diffraction peaks labeled as $q^S_i$, where the upper index $S$ stands for "slow cooling" and $i = 1, 2…$ indicates the cardinal number of the peaks at increasing $q$ values. The $i = 1–9$ peaks can be indexed as (11), (1$\bar{1}$), (02), (21), (03), (22), (13), and (3$\bar{2}$) with oblique lattice parameters of $a = 51.9$ Å, $c = 66.2$ Å, and $\beta = 85°$. The actual symmetry group is monoclinic $p2/m$ (i.e., an oblique columnar phase[19] rather than any of the other reported B1 phase structures[20, 21]). This Col$_{ob}$ phase corresponds to the B$^{obl}_{1Rev}$ phase, which is one of the three main columnar phases of bent-core molecules. This phase consists of tilted polar smectic ribbons with positions shifted periodically by half molecular lengths and polarizations alternating in and out along the ribbons. This phase along with the other two main columnar phases (B$_1$ and B$_{1Rev}$) is illustrated in Supplementary Figure 17. A non-tilted columnar phase (i.e. Col$_r$ or B$_{1Rev}$) was observed for **III**, and the (11) and (02) peaks appeared at higher values (around 0.21 Å⁻¹ and 0.3 Å⁻¹), indicating about 30% larger lattice parameters for **1** than for **III**. Additionally, the FWHM values of **1** are an order of magnitude larger than for **III**, showing short correlated columns (correlation length ~15 nm).

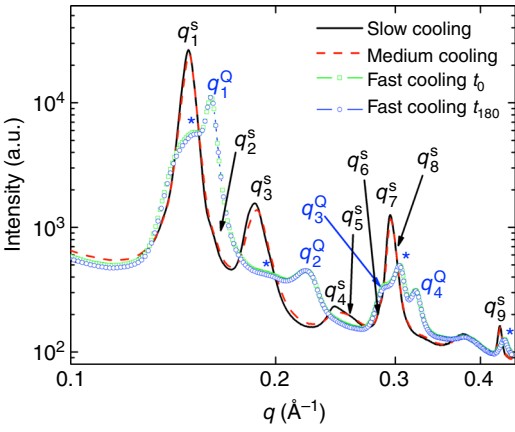

**Fig. 5** Small angle X-ray scattering (SAXS) analysis. Intensity *vs.* wave vector $q$ of compound **1**. Slow cooling: 5 °C min$^{-1}$; medium cooling: 8 °C min$^{-1}$; fast cooling: rapid quenching to room temperature from 200 °C. The position of the peaks: $q^S_1 = 0.1474$ Å$^{-1}$, $q^S_2 = 0.1610$ Å$^{-1}$, $q^S_3 = 0.1907$ Å$^{-1}$, $q^S_4 = 0.2428$ Å$^{-1}$, $q^S_5 = 0.2530$ Å$^{-1}$, $q^S_6 = 0.2854$ Å$^{-1}$, $q^S_7 = 2q^S_1 = 0.2956$ Å$^{-1}$, $q^S_8 = 0.3003$ Å$^{-1}$, and $q^S_9 = 0.4254$ Å$^{-1}$, $q^Q_1 = 0.1609$ Å$^{-1}$, $q^Q_2 = 0.2212$ Å$^{-1}$, $q^Q_3 = 0.2907$ Å$^{-1}$, $q^Q_4 = 2q^Q_1 = 0.3211$ Å$^{-1}$. Additional peaks in the SAXS pattern for the fast-cooled samples marked with blue * are at $q = 0.1519$ Å$^{-1}$, 0.1930 Å$^{-1}$, 0.3048 Å$^{-1}$, and 0.4336 Å$^{-1}$

The $q$ dependences of the SAXS signal for the quenched sample are shown right after quenching ($t = 0$) and 3 h later ($t = 180$ min). Both the fresh ($t = 0$) and relaxed ($t = 3$ h) quenched samples look identical indicating long stability. They represent peak positions $q^Q_i$ ($Q$ stands for "Quenched") different from that of the slow-cooled sample. The shoulder at lower $q$ (at 0.193 Å$^{-1}$ to the left of $q^Q_1 = 0.1609$ Å$^{-1}$ and marked with a blue *) is overlapping with $q^S_1$, and $q^Q_3 = 0.2907$ Å$^{-1}$, $q^Q_4 = 2q^Q_1 = 0.3211$ Å$^{-1}$ are also overlapping with $q^S_7$ and $q^S_8$. This indicates that the quenched sample is a combination of structural features inherent to the slow-cooled phase (see additional peaks for the quenched samples marked with a blue *) and another phase with $q^Q_1 = 0.1609$ Å$^{-1}$, $q^Q_2 = 0.2212$ Å$^{-1}$, $q^Q_3 = 0.2907$ Å$^{-1}$ and $q^Q_4 = 2q^Q_1 = 0.3211$ Å$^{-1}$ peaks. The presence of features inherent to the slow-cooled phase is probably due to the insufficient speed of quenching. The fact that the peak at $q^Q_2$ becomes a minority peak compared to $q^Q_1$ and $q^Q_4 = 2q^Q_1$ suggests that the second phase has a modulated layer structure similar to that of the B$_7$ phase (see Supplementary Note 9, Supplementary Figure 18) but with a short correlation length similar to the HNF$_{mod2}$ phase of material **II**. The recorded 2D patterns and fits are in Supplementary Note 10, Supplementary Figure 19 and Supplementary Figure 20.

**Scanning electron and atomic force microscopy**. We also performed SEM (Fig. 6) and AFM (Fig. 7) to study the self-assembly of the molecules in each phase obtained by slow and rapid cooling from the isotropic liquid state. The quenched sample of **1** clearly shows twisted filaments, which unlike typical HNFs, such as for compound **II**, appear more like corkscrews and have overall much larger dimensions (see Supplementary Note 11, Supplementary Figure 21 for a comparison of SEM images of **1** and **II**). For example, the width of the 'corkscrews' is about 250–300 nm in comparison to the 40 nm width of the filaments commonly reported for non-confined HNFs[12] such as compound **II**[11]. Due to the fact that the feature dimensions of these filaments far exceed the 100 nm size regime often established as the size regime for "nano", such filaments cannot be labeled as helical *nano*-filaments, but rather as helical micro-filaments (HµF).

A possible secondary twist of these filaments, however, is difficult to determine from these SEM images. Conceivably, the

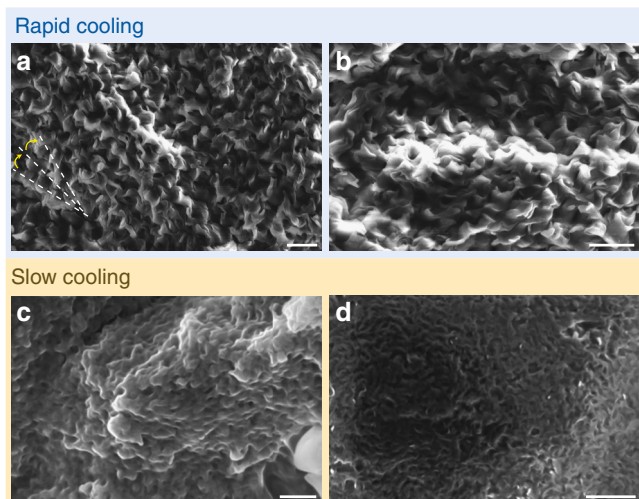

**Fig. 6** Scanning electron microscopy. SEM images of compound **1** obtained: **a**, **b** on quenching from the isotropic liquid state to room temperature at two different magnifications of the same sample and **c**, **d** on slow cooling at a rate of 5 °C min$^{-1}$ from the isotropic liquid phase to room temperature. Each sample (thickness about 200 µm) was imaged directly; i.e. no metal was deposited prior to imaging. Additional SEM images can be found in Supplementary Note 11, Supplementary Figure 21 and Supplementary Figure 22. Scale bars: 1 µm

bottom left section in Fig. 6a seems to show an area of the sample, where a right-handed secondary twist of filaments can be seen. On closer inspection of these images (Fig. 6a, Fig. 6b, Supplementary Figure 22a, and Supplementary Figure 22b), all filaments feature a right-handed twist, which is consistent with the observation of exclusive formation of right-handed HNFs by the parent homochiral compound **II**, as determined previously by TEM imaging[11]. The helical pitch is also much larger (about fivefold) for the HµF phase of the rapidly cooled compound **1** than for previously reported HNF phases (about ~900 nm–1 µm vs. the typically reported value of ~200 nm). The diameter and pitch of the filaments are in fact in between the HNF (B4) phase and the telephone wire (and often corkscrew) structure of the B7 phase of bent-core materials[22], which have modulated smectic layer structure[23, 24]. On the contrary, images of the slow-cooled sample do not show discrete filaments (Fig. 6c, d). The images almost appear to indicate an onset of filament formation, but an appropriate description could be the assembly of wavy layers forming some sort of sponge. The appearance is also not consistent throughout the sample (see Supplementary Note 11, Supplementary Figure 22c and Supplementary Figure 22d) unlike for the quenched sample of **1**, where all areas are identical.

We also obtained tapping mode atomic force microscopy (AFM) for both the slow cooled and the quenched sample of **1** (Fig. 7). The data for the Col$_{ob}$ phase seem to be more detailed than the SEM data shown in Fig. 6. In Fig. 7b, one can clearly see domains with columns oriented in different directions and the columns are aligned parallel to the surface. The height, amplitude, and phase shown in Fig. 7d–f, respectively, show images that are increasingly feature-rich. The phase image best reflects the chemical environment or nature on the surface (Fig. 7f). The cross-sectional analysis of selected regions plotted in Fig. 7a, c shows the modulation perpendicular to the columns. On the basis of the phase image, two periodicities were found $d_1 = 4.6$ nm and $d_2 = 3.5$ nm, which closely match values obtained from the wave vectors $q^S_1$ and $q^S_3$, respectively, measured by SAXS. The first

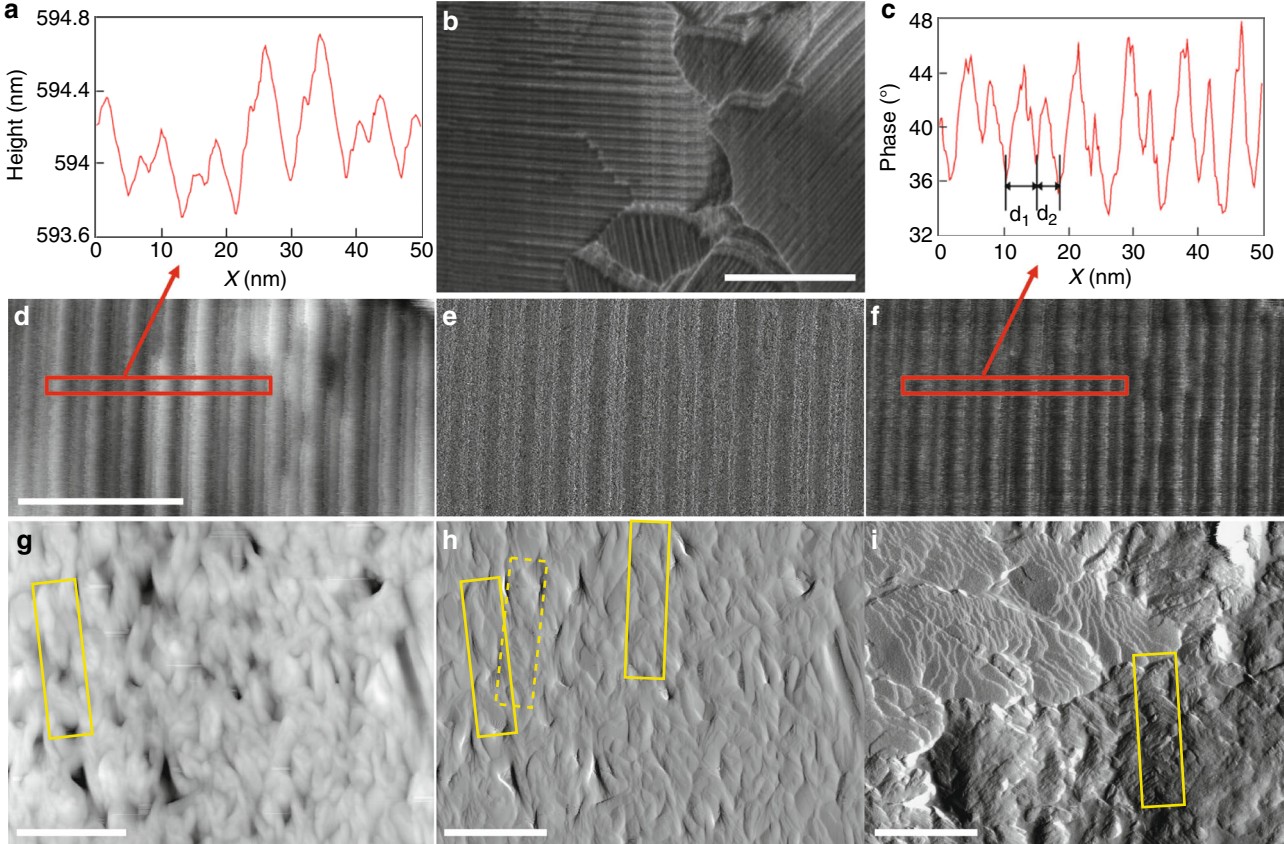

**Fig. 7** Tapping mode AFM images and image analysis of the $Col_{ob}$ phase obtained on slow cooling. **a** Cross-section analysis of selected area in **d**, **b** tapping mode AFM image, **c** cross-section analysis of selected area in **f**, **d**–**f** height, amplitude, and phase image, respectively. Tapping mode AFM images and image analysis of the HμF phase obtained on quenching from the isotropic liquid phase: **g** height, **h** and **i** amplitude. Isolated microfilaments are highlighted in yellow. Scale bars: in **b** 100 nm, in **d** for images **d**–**f** 40 nm, and in **g**–**i** 500 nm

periodicity arises from columns in the first plane, the second periodicity are partially hidden columns in the second plane.

On the contrary, the AFM images for the quenched sample (Fig. 7g–i) provide less detailed information in comparison to the SEM images shown in Fig. 6. Nevertheless, both height and amplitude data show some clearly discernable microfilaments with an average width as well as helical pitch matching the dimensions deduced from the SEM images (~200 width and ~850 nm helical pitch). The amplitude data shown in Fig. 7h appears to even show two microfilaments overlapping with a right-handed twist with the two neighboring filaments offset by about the same margin (about a quarter pitch length) as the highlighted filaments in the SEM image shown in Fig. 6a.

We also examined contact preparations between **1** and **II**, **1** and **III**, as well as **II** and **III**. Each contact sample was heated to a temperature where both compounds were in the isotropic liquid phase and then either cooled at a slow rate of 2 °C min$^{-1}$ or rapidly cooled (Supplementary Note 12, Supplementary Figure 23 - Supplementary Figure 27). In each case, a new phase appears to be induced at the contact zone as indicated by the birefringent texture occurring between the less birefringent textures of the two neat compounds. This is particularly pronounced for the contact preparation between **1** and **III** (Supplementary Figure 23 and Supplementary Figure 24). The most remarkable observation was made for the rapidly cooled contact preparation between **1** and **II** (Supplementary Figure 25). Here two unichiral HNF phases (HμF and HNF$_{mod2}$), composed of filaments that vastly differ in dimensions and helical pitch, meet and induce a texture with

conical features (Supplementary Figure 26) commonly observed for columnar phases of bent core LCs[16]. A similar observation was also made in the contact zone between compounds **II** and **III** (Supplementary Figure 27).

## Discussion

From a molecular point of view, compound **1** combines the left half of compound **II** with the right half of compound **III**. Both **II** and **III** form some variant of HNF phases (HNF$_{mod2}$ and HNF$_{mod}$, respectively), and the helical nanofilaments in both cases feature the same dimensions, the same helical pitch, and only differ in their formation of a unichiral HNF phase *vs.* an HNF phase conglomerate. One would assume that the homo-chirality of **II** is critical for the formation of the unichiral HNF$_{mod2}$ phase, where the only additional degree of ordering is the interlayer modulation not shared by the HNF$_{mod}$ phase formed by **III**. We have previously also shown that a racemic mixture of compounds related to **II** with two racemic side chains (i.e., (*R, R*), (*S,S*), (*R,S*) and (*S,R*)) does not form an HNF phase, but rather short correlated crystalline ribbons with an overall columnar structure[11]. Astonishingly, the combination of the two molecular halves of **II** and **III** in compound **1** does not only lead to a cooling rate-dependent formation of entirely different phases, but the HNF phase formed on rapid cooling leads to differently shaped helical filaments (similar to rotini noodles or corkscrews) with much larger overall dimensions, as well as an order of magnitude larger helical pitch than any other

HNF phase material (Supplementary Note 13, Supplementary Figure 28).

Considering the phase sequence of all three compounds shown in Fig. 1, it seems that the right half of **1** (the *para*-side) desires to form the thermodynamically more stable columnar B1 phase similar to compound **III** (Col$_{ob}$ vs. Col$_r$), whereas the left half (the *meta*-side) promotes the kinetically more stable filament-type B4 phase, as found in compound **II**. On the basis of these observations, we envision to find more examples in anisotropic bent-core molecules, where different sides of the molecule contain molecular features of symmetric bent-core molecules that have distinctly different phase sequences. Such hypothesis of course is subject to further studies. The textures observed in contact preparations at the interfaces between **1** and **II**, **1** and **III**, and between materials **II** and **III** also demonstrate the subtle differences of the free energies between the Col$_{ob}$ (or Col$_r$), HNF and HμF phases of these bent-core liquid crystals.

In conclusion, the unprecedented formation of two liquid crystalline polymorphs from the isotropic liquid state under different thermal history, clearly demonstrates that these structurally different phases have almost identical free energies, so their formation can be selected by small differences in thermal history or concentrations. Astonishingly, such a small group of structurally very closely related compounds displays a very rich variety of self-assembled chiral superstructures including helical nanofilament and the heretofore never reported helical microfilament phases.

## Methods

**Optical characterization**. Polarized optical microscopy (POM) observations were carried out under an Olympus BX-53 polarizing microscope equipped with a Linkam LTS420E heating/cooling stage.

**Spectroscopy**. Thin film circular dichroism (CD) spectropolarimetry was performed using an OLIS spectrophotometer with quartz substrates. UV-Vis transmission and reflection spectra were measured using either a USB-4000 (Ocean Optics) or a Lambda 18 (Perkin Elmer) UV-Vis spectrophotometer.

**SAXS**. SAXS was carried out on beamline 7.3.3 of the Advanced Light Source of Lawrence Berkeley National Laboratory[25] (10 keV incident beam energy, 1.24 Å wavelength, utilizing a Pilatus 2 M detector). The materials were filled into 1 mm diameter quartz X-ray capillary tubes, which were then mounted into a custom-built aluminum cassette that allowed X-ray detection with ± 13.5° angular range. The cassette fits into a standard hot stage (Instec model HCS402) that allowed temperature control with ± 0.1 °C precision. The stage also included two cylindrical neodymium iron boron magnets that supplied a magnetic induction of $B = 1.5$ T perpendicular to the incident X-ray beam. The analysis were proceed in Igor Pro software with Nika package[26].

**Scanning electron and atomic force microscopy**. Scanning electron microscopy (SEM) analysis was performed using a Quanta 450 FEG SEM. Atomic force microscopy (AFM) images were taken in tapping mode at room temperature at the Molecular Foundry at the Lawrence Berkeley National Laboratory. We utilized Si probes coated with Al, $k = 5$ N/m, $f = 150$ kHz.

**Data availability**. The authors declare that the data supporting the findings of this study are available within the paper and its supplementary information file. Any additional information that supports the findings of this study is available from the corresponding authors upon reasonable request.

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

## Acknowledgements

This work was supported by the U.S. National Science Foundation (NSF, DMR-1506018 and DMR 1307674), the Ohio Third Frontier (OTF) program for Ohio Research Scholars "Research Cluster on Surfaces in Advanced Materials" (T.H.), which also supports the Liquid Crystal Characterization facility at the Liquid Crystal Institute (Kent State University), where current SEM data were acquired. We are also grateful for access to the beamline 7.3.3 at the Advanced Light Source and to the AFM at the Molecular Foundry at the Lawrence Berkeley National Laboratory, which are supported by the Director (Office of Science, Office of Basic Energy Sciences) of the U.S. Department of Energy under Contract No. DE-AC02-05CH11231. M.S. and C.Z. thank Paul Ashby in the Molecular Foundry (LBNL) for the help with AFM measurements; S.S. and T.H. wish to thank Dr. Oleg D. Lavrentovich for access to the UV-vis transmission and reflection setups.

## Author contributions

T.H. and A.J. directed the research; L.L. performed the synthesis and chemical characterization as well as the POM, DSC, CD, and SEM studies. S.S. performed additional DSC and UV-Vis reflection and transmission experiments. C.Z. and M.S. performed the

SAXS studies and M.S. the AFM imaging. T.H. and A.J., with contributions from all co-authors, wrote the manuscript.

## Additional information

**Competing interests:** The authors declare no competing financial interests.

