## [Peer Review File · Nature Communications]

Reviewers' comments:

Reviewer #2 (Remarks to the Author):

The manuscript by Lin Li et al. describes the observation of thermal-induced structural polymorphism in a bent-core liquid crystal compound. This is the first time that thermal-induced structural polymorphism has been observed in a thermotropic liquid crystal compound. Definitive quantitative X-ray and DSC measurements are presented supported by POM, SEM, CD, and AFM measurements. The novelty of this unambiguous observation of thermal-induced structural polymorphism in a bent-core liquid crystal compound warrants publication in Nature Communications. The description of each measurement contains adequate details for the measurements to be repeated by others. The discovery will impact the development of roll-to-roll film printing. It was recently demonstrated that in the rapid film printing from a rectangular capillary, the fluid liquid crystal/solvent mixture in the capillary undergoes an intermediate mesophase before solidifying into a printed semiconducting film [1]. The possibility of kinetic-trapping of different polymorphs could enable better tuning of the properties of the printed film.

Nonetheless, prior to publication, several points require further clarification.

1. Although this is the first observation of thermal-induced structural polymorphism in liquid crystals, there has been a previous observation of electric-field-induced structural polymorphism with the zero-field structure being pathway dependent [2]. This discovery involved a rod-shaped liquid crystal compound and should be mentioned in the publication since it is relevant to the topic.
2. The descriptions of the SAXS peak positions for the quenched sample are inconsistent between the main text and the caption for Figure 5. Also, there appears to be a weak peak at $\sim 1.9 \text{ \AA}^{-1}$ (comparable in intensity to the qS2 peak) that is neglected. The inconsistencies between text and figure caption should be fixed and the reason for neglecting the weak peak explained.
3. There are at least 3 variants of the modulated B1 phase [3-5]. All of these variants exhibit the same SAXS scattering pattern and other considerations need to be taken into account to distinguish between them. It wasn't clear which B1 structure the authors were claiming for their crystal polymorph phase.

[1] J. Wan et al., APL Mater. 4, 016103 (2016).

- [2] S. Jaradat et al., Appl. Phys. Lett. 94, 153507 (2009).
- [3] J. Szydłowska et al., Phys. Rev. E 67, 031702 (2003).
- [4] C.L. Folcia et al., Phys. Rev. E 74, 031702 (2006).
- [5] J. Martínez-Perdiguero et al., Phys. Rev. E 82, 041706 (2010).

Reviewer #3 (Remarks to the Author):

The manuscript reports for the first time very interesting results concerning polymorphism in a liquid crystal. Depending on the rate of cooling from the isotropic liquid state two structurally and morphologically noticeably different mesophases are formed. Additionally a new type of helical filament liquid crystalline organization for a bent-core compound is described. Furthermore, the choice of cooling rate also provides a different structural color, which could be exploited and joined to claimed practical applications for this type of chiral compact molecular packing.

Results and proposals are supported by appropriate and consistent experimental data from a variety of techniques, to point POM, CD, SAXS, SEM and AFM studies.

The paper is clearly written and has a good, legible structure.

The used synthesis procedures are quite well established and seem to be correctly followed. Experimental results, in general, are accurately presented and discussed and conclusions are supported by the presented measurements.

This paper contributes to an ongoing investigation in the community and, presents a valuable dataset. I find the topic and the results of the paper important even for a wide public and not only the liquid crystal researchers. If published, will trigger further experimental, computational, and theoretical works. They do afford significant advance and novelty and I recommend this paper to be published in Nature Communication once the authors consider the points given below, which may be useful to improve the clarity of this manuscript for general audience.

- All Figures from the Supporting Information should be clearly identified and referenced at the main text.

- Authors well-describe the thermal behavior of compounds II and III on further cooling, and report that glassy mesophases occurred.

However, in my opinion, this is not the case for compound 1 description.

From Figure 2 data, in contrast to compounds II and III, can be deduced that compound 1 never form crystalline solids or glassy mesophases on further cooling or keeping the samples at room temperature?

Have the authors explored the stability of both mesophases (columnar and helical filaments) for periods of time longer than 3 hours?

In my opinion, all these aspects should be clearly checked and detailed in the manuscript.

- Table S1 shows data from DSC studies from a unique sample after different thermal treatments (pristine sample, fast cooled and slow cooled sample from liquid) or do they correspond to different samples?

What do 2nd heating runs means? A 2nd heating once the samples was cooled inside the oven?

I got some doubts and comments.

According to the Figure S4 captions, results from second heating runs for A and B should not be similar?. In both cases, samples are cooled at 5 °C / min from liquid. Why results are different?

Furthermore, second heating run for C is different to those for A and B as 50 °C/ min heating rate was used. But why do the authors heat at this rate?

At the main text very simple comments are given concerning DSC data but somehow they are not full consistent and supported with those reported at the Supporting Information.

- Also related with DSC data.

Pristine compound 1 from ethanol, does form a crystalline solid?

From Figure 4 and comments through the main text it seems a solid.

Is this consistent with data from DSC (A)? First and second heating runs of A deal with the same aggregation state?

What about XRD, CD, SEM ... data of pristine samples of compound 1?

In my opinion, all these aspects are not really clear in the manuscript.

- Comments about color differences are poor. Could the authors afford comments and/or UV-vis data to support the origin for color differences?

- Discussions concerning B7 phase is hard to follow even more without schemes and photos. Please insert some cartoon as supporting information.

- For a better reading, Figure 3 should include temperatures as part of the photos.

- In the case of comment: "Conceivably the bottom left section in Figure 6A seems to show an area of the sample, where a right-handed secondary twist of filaments can be seen." , this area should be identified properly.

- The goal and interest of contact preparation studies and results are not really clear.

- Authors have included very attractive and clear models of the different helical filament packing mentioned in the manuscript. However, this is not the case for the columnar organizations. For clarity illustrations of these molecular arrangements are welcome.

Furthermore, have the authors any suspicion of possible structural relationship and molecular packing between both mesophases? Do they have a common origin?

- AFM provides interesting support for the columnar organization. What about AFM studies with the helical filaments?

- Authors discuss the formation of polymorphs and different molecular packing in terms of subtle intermolecular interactions depending on the molecular structure.

But is this just a solely example or could these results be extended to further bent-core structures? Do the authors have further results as to expect this sort of polymorphism for other molecular designs?

From previous reported results referenced by the authors, the formation of HNF is trending for bent-core molecules containing several biphenyl-units. Have this type of molecules "special coined" reasons to induce helical filament-like aggregations?

- Some minor comment about experimental synthesis:

- Do yields make sense with decimals?

- Authors state that compound 1 and compound 4 have been synthesized using similar reaction conditions, however yields were very different. Do the authors have any explanation?

Point-by-point response (blue font) to the referees' comments

Reviewer #2 (Remarks to the Author):

The manuscript by Lin Li et al. describes the observation of thermal-induced structural polymorphism in a bent-core liquid crystal compound. This is the first time that thermal-induced structural polymorphism has been observed in a thermotropic liquid crystal compound. Definitive quantitative X-ray and DSC measurements are presented supported by POM, SEM, CD, and AFM measurements. The novelty of this unambiguous observation of thermal-induced structural polymorphism in a bent-core liquid crystal compound warrants publication in Nature Communications. The description of each measurement contains adequate details for the measurements to be repeated by others. The discovery will impact the development of roll-to-roll film printing. It was recently demonstrated that in the rapid film printing from a rectangular capillary, the fluid liquid crystal/solvent mixture in the capillary undergoes an intermediate mesophase before solidifying into a printed semiconducting film [1]. The possibility of kinetic-trapping of different polymorphs could enable better tuning of the properties of the printed film.

Nonetheless, prior to publication, several points require further clarification.

1. Although this is the first observation of thermal-induced structural polymorphism in liquid crystals, there has been a previous observation of electric-field-induced structural polymorphism with the zero-field structure being pathway dependent [2]. This discovery involved a rod-shaped liquid crystal compound and should be mentioned in the publication since it is relevant to the topic.

Thank you for your favorable assessment of our work and to draw our attention to the significance of these results in film printing. We also thank for drawing our attention to a previous field-induced polymorphism. We have included a statement about this effect reported by H. Gleeson and provided the reference.

2. The descriptions of the SAXS peak positions for the quenched sample are inconsistent between the main text and the caption for Figure 5. Also, there appears to be a weak peak at $\sim 1.9 \text{ \AA}^{-1}$ (comparable in intensity to the q_{S2} peak) that is neglected. The inconsistencies between text and figure caption should be fixed and the reason for neglecting the weak peak explained.

We have corrected the inconsistencies between text and figure caption. We believe that by the $\sim 1.9 \text{ \AA}^{-1}$ peak the referee means the plateau (shoulder) at 0.193 \AA^{-1} , between q_1^O and q_2^O and right below q_3^S . This overlap further reinforces our statement that the "the quenched sample is a combination of structural features inherent to the slow cooled phase." We have highlighted this particular peak also with an asterisk and included this q value in the figure caption.

3. There are at least 3 variants of the modulated B1 phase [3-5]. All of these variants exhibit the same SAXS scattering pattern and other considerations need to be taken into account to distinguish between them. It wasn't clear which B1 structure the authors were claiming for their crystal polymorph phase.

[1] J. Wan et al., APL Mater. 4, 016103 (2016).

[2] S. Jaradat et al., Appl. Phys. Lett. 94, 153507 (2009).

[3] J. Szydłowska et al., Phys. Rev. E 67, 031702 (2003).

[4] C.L. Folcia et al., Phys. Rev. E 74, 031702 (2006).

[5] J. Martínez-Perdiguero et al., Phys. Rev. E 82, 041706 (2010).

We agree that in addition to the originally called B1 phase, which has an overall antiferroelectric structure with $p2mg$ symmetry, there were three other columnar phases described: two phases (so called B1rev and B1rev phase with splayed polarization) are non-tilted, and one tilted (B1rev tilted) .

Liquid Crystal Institute
Chemical Physics Interdisciplinary Program
P.O. Box 5190 • Kent, Ohio 44242-0001

330-672-2654 • Telefax: 330-672-2796 • E-mail: mail@lci.kent.edu • www.lci.kent.edu

The space group ($p2/m$), which we already have given in the original submission, corresponds to an oblique columnar phase. In the revised paper we have clarified this and have provided the suggested references (new references [19-21]).

Reviewer #3 (Remarks to the Author):

The manuscript reports for the first time very interesting results concerning polymorphism in a liquid crystal. Depending on the rate of cooling from the isotropic liquid state two structurally and morphologically noticeably different mesophases are formed. Additionally a new type of helical filament liquid crystalline organization for a bent-core compound is described. Furthermore, the choice of cooling rate also provides a different structural color, which could be exploited and joined to claimed practical applications for this type of chiral compact molecular packing.

Results and proposals are supported by appropriate and consistent experimental data from a variety of techniques, to point POM, CD, SAXS, SEM and AFM studies.

The paper is clearly written and has a good, legible structure.

The used synthesis procedures are quite well established and seem to be correctly followed. Experimental results, in general, are accurately presented and discussed and conclusions are supported by the presented measurements.

This paper contributes to an ongoing investigation in the community and, presents a valuable dataset. I find the topic and the results of the paper important even for a wide public and not only the liquid crystal researchers. If published, will trigger further experimental, computational, and theoretical works. They do afford significant advance and novelty and I recommend this paper to be published in Nature Communication once the authors consider the points given below, which may be useful to improve the clarity of this manuscript for general audience.

- All Figures from the Supporting Information should be clearly identified and referenced at the main text.

We double-checked and all figures from the Supplementary Information are now mentioned in the main text.

- Authors well-describe the thermal behavior of compounds II and III on further cooling, and report that glassy mesophases occurred.

However, in my opinion, this is not the case for compound 1 description.

This might be a misunderstanding. We have not made a specific comment about compound **1** forming a glassy state. To make it clear in the revised version we have included a sentence that highlights that **1** does not appear to form a glass. Even after many days standing at room temperature, no glass formation or crystallization was observed. DSC data are in agreement with this statement.

From Figure 2 data, in contrast to compounds II and III, can be deduced that compound 1 never form crystalline solids or glassy mesophases on further cooling or keeping the samples at room temperature? Have the authors explored the stability of both mesophases (columnar and helical filaments) for periods of time longer than 3 hours?

In my opinion, all these aspects should be clearly checked and detailed in the manuscript.

We have included a statement that there is no glass formation and that the two phases appear to be stable for longer period of time without crystallization. Both phases formed on cooling are indeed monotropic. POM clearly reveals that the pristine sample stays crystalline until the first transition to the isotropic liquid phase at 153 °C.

Liquid Crystal Institute
Chemical Physics Interdisciplinary Program
P.O. Box 5190 • Kent, Ohio 44242-0001

330-672-2654 • Telefax: 330-672-2796 • E-mail: mail@lci.kent.edu • www.lci.kent.edu

- Table S1 shows data from DSC studies from a unique sample after different thermal treatments (pristine sample, fast cooled and slow cooled sample from liquid) or do they correspond to different samples? What do 2nd heating runs mean? A 2nd heating once the samples were cooled inside the oven? I got some doubts and comments.

We included several clarifying statements in the bottom caption of Table S1 and as well as in the captions of the three DSC traces (Figure S4).

According to the Figure S4 captions, results from second heating runs for A and B should not be similar?. In both cases, samples are cooled at 5 °C / min from liquid. Why results are different?

We believe that both DSC experiments show almost identical data, particularly on heating. The cooling runs show peak transition temperatures that are ~3 °C lower in plot (B) vs. plot (A). One possible explanation could be some trapped solvent or simply some instrumental inconsistencies (e.g., flow of nitrogen). Another explanation could be that some molecular associations from the microfilaments persist well into the isotropic liquid phase and that this causes this slight discrepancy between the cooling runs – this sample was rapidly cooled before the DSC measurements were performed, i.e. the H_μF phase was present.

Furthermore, second heating run for C is different to those for A and B as 50 °C/ min heating rate was used. But why do the authors heat at this rate?

We used this higher heating/cooling rate to simulate the formation of the H_μF phase (thermal quench), and while this rate is not sufficiently high for complete formation of the B4 microfilament phase, some B4 should be present at this rapid cooling rate. While the peak phase transition temperatures are again basically identical to the other two samples (plots (A) and (B)), the peak phase transition temperatures now occur at even lower temperatures.

At the main text very simple comments are given concerning DSC data but somehow they are not full consistent and supported with those reported at the Supporting Information.

We believe that the structural, optical, spectral and morphological data discussed in more detail in the main manuscript are more significant in supporting this unique polymorphism observed for compound **1**.

- Also related with DSC data.

Pristine compound 1 from ethanol, does form a crystalline solid?

Yes, the pristine sample does form a crystalline solid and POM clearly reveals that the two phases formed on slow or rapid cooling are monotropic in the first heating run (both by DSC and POM). The really surprising fact is that there is no sign of crystallization neither by POM nor by DSC. The phases are stable for days (weeks).

From Figure 4 and comments through the main text it seems a solid.

Is this consistent with data from DSC (A)? First and second heating runs of A deal with the same aggregation state?

Yes, consistent with DSC data. We have included here a set of three DSC runs from a second instrument (three heating/cooling runs). There is only one peak on heating and one peak on cooling and POM confirms that the first heating shows the melting of the compound at 153 °C. These data are not included in the Supplementary Information, because they would duplicate existing datasets. While the aggregation state is

Liquid Crystal Institute
Chemical Physics Interdisciplinary Program
P.O. Box 5190 • Kent, Ohio 44242-0001

330-672-2654 • Telefax: 330-672-2796 • E-mail: mail@lci.kent.edu • www.lci.kent.edu

not the same, it shows how energetically close the polymorphs are, including the crystalline phase formed by the pristine sample of **1**.

What about XRD, CD, SEM ... data of pristine samples of compound **1**?
In my opinion, all these aspects are not really clear in the manuscript.

Our polarized light optical microscopy observations clearly support the presence of the CoI_{ob} ($B1_{rev\ tilt}$) phase for the pristine sample (recrystallized from ethanol), and on slow cooling from the isotropic liquid phase.

- Comments about color differences are poor. Could the authors afford comments and/or UV-vis data to support the origin for color differences?

We have now performed additional UV-vis reflection and transmission experiments (see Supplementary Section 8 Supplementary, Supplementary Figure 6), which confirm that the blue color of the thermally quenched sample appears in reflection (structural color). In the transmission experiments for this same sample scattering of the sample leads to very broad transmission with the highest transmission intensity at wavelengths complementary to the blue reflection color.

- Discussions concerning B7 phase is hard to follow even more without schemes and photos. Please insert some cartoon as supporting information.

We have included a drawing (specify Supplementary Figure 17B) and a reference relevant for the B7 phase.

- For a better reading, Figure 3 should include temperatures as part of the photos.

Liquid Crystal Institute
Chemical Physics Interdisciplinary Program
P.O. Box 5190 • Kent, Ohio 44242-0001

330-672-2654 • Telefax: 330-672-2796 • E-mail: mail@lci.kent.edu • www.lci.kent.edu

It appears that excessive numbering within Figures should be avoided for the journal, so we retained the temperatures in the Figure Caption.

- In the case of comment: "Conceivably the bottom left section in Figure 6A seems to show an area of the sample, where a right-handed secondary twist of filaments can be seen.", this area should be identified properly.

We have adjusted Figure 6A and now show the area where a secondary twist can be seen.

- The goal and interest of contact preparation studies and results are not really clear.

The goal is to show that all these different B4 phases are morphologically and structurally unique and in support of this statement these phases do not mix in the contact zone. Future, more intensive studies on the phases present in the contact zones will be reported in due course.

- Authors have included very attractive and clear models of the different helical filament packing mentioned in the manuscript. However, this is not the case for the columnar organizations. For clarity illustrations of these molecular arrangements are welcome.

The Col_{ob} phase we observed corresponds to the B^{obl}_{1Rev} phase, which is one of the three main columnar phases formed by bent-core molecules. As we described in the text, it has a $p2/m$ symmetry whereby tilted polar smectic ribbons with positions shifted periodically by half molecular lengths and polarizations alternating in and out along the ribbons. This phase along with the other two main columnar phases (B₁ and B_{1Rev}) is illustrated in the Figure S18 of the Supplementary Information.

Furthermore, have the authors any suspicion of possible structural relationship and molecular packing between both mesophases? Do they have a common origin?

I think this is the argument we are trying to make – energetically it seems the two phases are close, one is formed under kinetic, the other one under thermodynamic control. The HNF_{mod}, HNF_{mod2} and H μ F phases have columnar local structures in a similar way as HNF has locally smectic structures. The helical filaments could be considered secondary structures and the smectic or columnar phases as primary structures. The primary structures determine the thermodynamically stable phase and the secondary structure is formed dynamically.

- AFM provides interesting support for the columnar organization. What about AFM studies with the helical filaments?

We have reported very detailed morphological data from SEM imaging for the H μ F phase both in the manuscript and the Supplementary Information. For the Col_{ob} phase, SEM appeared not very useful, which is why we performed additional AFM studies. We have now also included images from AFM investigations of the microfilament phase (Figures 7E-G).

- Authors discuss the formation of polymorphs and different molecular packing in terms of subtle intermolecular interactions depending on the molecular structure.

But is this just a solely example or could these results be extended to further bent-core structures? Do the authors have further results as to expect this sort of polymorphism for other molecular designs?

Unfortunately it is difficult to devise any design rules for bent core molecules showing this type of behavior, since this is the only example for which such drastic behavior has been reported to date. Based on these observations we envision to find more examples in anisotropic bent-core molecules where different sides of

Liquid Crystal Institute
Chemical Physics Interdisciplinary Program
P.O. Box 5190 • Kent, Ohio 44242-0001

330-672-2654 • Telefax: 330-672-2796 • E-mail: mail@lci.kent.edu • www.lci.kent.edu

the molecule contain molecular features of symmetric bent-core molecules that have distinctly different phase sequences. Such hypothesis of course is subject to further studies.

Reported results referenced by the authors, the formation of HNF is trending for bent-core molecules containing several biphenyl-units. Have this type of molecules “special coined” reasons to induce helical filament-like aggregations?

Maybe. The tris-biphenyl derivatives seem prone to B4 phase formation, and considering the proposed mechanism of formation – i.e. the frustrated desire for orthogonal layer growth of bottom and top halves of these types of molecules leading to local saddle splay, could be favored by molecules with more extensive rigid aromatic arms. Yet, there are examples of B4 phase formation for more “traditional” bent-core molecules lacking the biphenyl arms.

- Some minor comment about experimental synthesis:
- Do yields make sense with decimals?

No, and we corrected that.

- Authors state that compound 1 and compound 4 have been synthesized using similar reaction conditions, however yields were very different. Do the authors have any explanation?

Yes, we have an explanation for that: Compound **4** features a THP protecting group, which is reportedly stable under DCC type esterification conditions. Nevertheless, we did observe significant deprotection and this de-protected side-product (actually compound **3**) was isolated and used. However, what we report here is the yield for the pure, isolated compound **4**, which is lower due to the indicated de-protection taking place during the reaction and reaction work-up.

Liquid Crystal Institute
Chemical Physics Interdisciplinary Program
P.O. Box 5190 • Kent, Ohio 44242-0001

330-672-2654 • Telefax: 330-672-2796 • E-mail: mail@lci.kent.edu • www.lci.kent.edu

Reviewers' Comments:

Reviewer #2:

Remarks to the Author:

The authors have thoughtfully and fully addressed the issues and appropriately modified and improved their original manuscript. I recommend publication without further delay.

Reviewer #3:

Remarks to the Author:

The manuscript has been improved significantly according to referee's comments and, from my point of view, it is suitable for publication in Nature Communications. However, first I suggest the authors to revise witting concerning DSC data comments.

it is not a strictly necessary requirement to consider all these points for the final acceptance of the manuscript, but it would improve the clarity of this part of the manuscript.

1.- Revised manuscript text and supplementary information:

1.a: "DSC measurements were performed of material 1 in three different conditions: (i)

recrystallized from ethanol (neat), (ii) after heating the sample to the isotropic liquid phase

followed by quenching, and (iii) after heating to the isotropic liquid phase and slow cooling at a rate of 5 °C_min⁻¹. The resulting thermograms obtained from processes (i) and (ii) are nearly identical in the first and second heating/cooling runs and only differ in the about 3 °C lower phase transitions recorded on cooling for the quenched sample (Supplementary Section 6, Supplementary Table 1 and Supplementary Figure 4A and Supplementary Figure 4B)."

1.b.: "Supplementary Figure 4. DSC plots for compound 1 directly taken from the Perkin Elmer Pyris1 software interface. (A) Sample was obtained directly from recrystallization and measured at 5 °C_min⁻¹. In this case, the sample forms the B1 (p2/m, Colob) phase based on POM observations. The phase formed is monotropic (i.e. only observed on cooling). (B) Sample was obtained by cooling

from the isotropic liquid phase and then thermally quenched (outside of the DSC instrument) and then measured at a rate of 5 °Cmin⁻¹ by DSC. In this case, the sample forms the B1 (p2/m, Colob) phase. (C) Sample was obtained by cooling from the isotropic liquid phase at 5 °Cmin⁻¹ (outside of the DSC instrument) and then measured by DSC at a rate of 50 °Cmin⁻¹. In this case, the sample initially forms the B1 (p2/m, Colob) phase, which should then coexist with the H_μF B4 phase during the 50 °Cmin⁻¹ cooling runs. It appears that the

broader phase transitions occurring at lower temperatures are a consequence of the 10-fold higher heating/cooling rate.”

Is (B) ok at the figure caption?

According to Supplementary Table 1, plot B is for an original sample heated to Iso and then quickly cooled in the air. Is the Col polymorph formed from a quick cooling?

2. Point 1.a and the point-by-point response:

“We have included here a set of three DSC runs from a second instrument (three heating/cooling runs. There is only one peak on heating and one peak on cooling and POM confirms that the first heating shows the melting of the compound at 153 °C. These data are not included in the Supplementary Information, because they would duplicate existing datasets. While the aggregation state is not the same, it shows how energetically close the polymorphs are, including the crystalline phase formed by the pristine sample of 1.”

From my point of view, authors could point out “how energetically close the polymorphs are, including the crystalline phase formed by the pristine sample of 1”. 2nd run heating from plot A corresponds to DSC data at 5 °Cmin⁻¹ rate of a sample heated to liquid phase then cooled at 5 °Cmin⁻¹ (i.e. a slowly cooled sample). Thus plots A and B, experiments carried out at 5 °Cmin⁻¹ allow to propose this.

This is really surprising and very interesting! ... Data from different techniques are consistent with this.

3. I do not completely agree with some authors’ comments.

a) “We believe that both DSC experiments show almost identical data, particularly on heating. The cooling runs show peak transition temperatures that are ~ 3 °C lower in plot (B) vs. plot (A). One possible explanation could be some trapped solvent or simply some instrumental inconsistencies (e.g., flow of nitrogen).”

These are not very consistent reasons. Have the authors repeated measurements?

“Another explanation could be that some molecular associations from the microfilaments persist well into the isotropic liquid phase and that this causes this slight discrepancy between the cooling runs – this sample was rapidly cooled before the DSC measurements were performed, i.e. the H μ F phase was present.”

It could be possible. There exists references of bent-core molecules where some molecular aggregation persists in the liquid phase.

b) Concerning results from 50 °C/min rate experiments.

“The sample of 1 treated according to process (iii), however, shows broader phase transitions with the phase transition on cooling now recorded about 11 to 14 °C lower than for the neat sample treated according to process (i). Process (iii) used a heating/cooling rate of 50 °C_{min}⁻¹ during the DSC experiment and a coexistence of the two phases observed on slow and rapid cooling, respectively, is expected (Supplementary Table 1 and Supplementary Figure 4C). All plots show just one phase transition peak on heating and cooling, indicative of a monotropic liquid crystal phase (only formed on cooling), with virtually equal phase transition enthalpies, but lower phase transition temperatures for each of the thermally-treated samples.”

- Experiments carried out with slow cooled samples but studied at different rates, at 50 °C/min and 5 °C/min should be well differentiated.

- Taking into account that Plot C concerns starting samples after a slow cooling from liquid (i.e. Col polymorph is formed), why did the authors expect the coexistence of the two phases on heating at a rate of 50 °C/min?

- Assuming that both polymorphs are energetically so close and both lead to liquid at similar temperatures, why do authors expect neat peaks for each potential transition?

- Broad peaks and sometimes, lower transition temperatures, can be expected on increasing rates in DSC measurements.

Could be completely ruled out the coexistence of both polymorphs on samples? Could this sort of coexistence lead to broad peak as it is very often observed for mixtures?

Point-by-point response (blue font) to the referees' comments - NCOMMS-17-24181A

Reviewer #2

The authors have thoughtfully and fully addressed the issues and appropriately modified and improved their original manuscript. I recommend publication without further delay.

No changes requested.

Reviewer #3

The manuscript has been improved significantly according to referee's comments and, from my point of view, it is suitable for publication in Nature Communications. However, first I suggest the authors to revise witting concerning DSC data comments.

it is not a strictly necessary requirement to consider all these points for the final acceptance of the manuscript, but it would improve the clarity of this part of the manuscript.

Is (B) ok at the figure caption?

According to Supplementary Table 1, plot B is for an original sample heated to Iso and then quickly cooled in the air. Is the Col polymorph formed from a quick cooling?

We have corrected this statement in the figure caption of the DSC plots in the Supplementary Information. It is indeed initially the microfilament phase. We have also added corrective statements in the main text.

From my point of view, authors could point out "how energetically close the polymorphs are, including the crystalline phase formed by the pristine sample of 1". 2nd run heating from plot A corresponds to DSC data at 5 oCmin-1 rate of a sample heated to liquid phase then cooled at 5 oCmin-1 (i.e. a slowly cooled sample). Thus plots A and B, experiments carried out at 5 oCmin-1 allow to propose this.

This is really surprising and very interesting! ... Data from different techniques are consistent with this.

We have now pointed this out explicitly on page 7 of the main manuscript (end of first paragraph).

a) "We believe that both DSC experiments show almost identical data, particularly on heating. The cooling runs show peak transition temperatures that are ~3 °C lower in plot (B) vs. plot (A). One possible explanation could be some trapped solvent or simply some instrumental inconsistencies (e.g., flow of nitrogen)."

These are not very consistent reasons. Have the authors repeated measurements?

All DSC measurements are reproducible. We have repeated these measurements several times and using multiple instruments (as stated and as per data shown in our previous Response to the Referees' Comments letter. Therefore, we might be in a position to actually rule out errors originating from trapped solvent or instrumental inconsistencies.

"Another explanation could be that some molecular associations from the microfilaments persist well into the isotropic liquid phase and that this causes this slight discrepancy between the cooling runs – this sample was rapidly cooled before the DSC measurements were performed, i.e. the H_μF phase was present."

Liquid Crystal Institute
Chemical Physics Interdisciplinary Program
P.O. Box 5190 • Kent, Ohio 44242-0001

330-672-2654 • Telefax: 330-672-2796 • E-mail: mail@lci.kent.edu • www.lci.kent.edu

It could be possible. There exists references of bent-core molecules where some molecular aggregation persists in the liquid phase.

We believe this to be the case, and we have stated this in the revised manuscript and highlighted this again on page 7 in the final revised manuscript (middle first paragraph).

b) Concerning results from 50 °Cmin⁻¹ rate experiments.

“The sample of 1 treated according to process (iii), however, shows broader phase transitions with the phase transition on cooling now recorded about 11 to 14 °C lower than for the neat sample treated according to process (i). Process (iii) used a heating/cooling rate of 50 °C_{min}⁻¹ during the DSC experiment and a coexistence of the two phases observed on slow and rapid cooling, respectively, is expected (Supplementary Table 1 and Supplementary Figure 4C). All plots show just one phase transition peak on heating and cooling, indicative of a monotropic liquid crystal phase (only formed on cooling), with virtually equal phase transition enthalpies, but lower phase transition temperatures for each of the thermally-treated samples.”

- Experiments carried out with slow cooled samples but studied at different rates, at 50 °Cmin⁻¹ and 5 °Cmin⁻¹ should be well differentiated.

We trust that the added comments on page 7 of the revised manuscript and in the Supplementary Information (page 14) will allow the reader to clearly differentiate these two datasets.

- Taking into account that Plot C concerns starting samples after a slow cooling from liquid (i.e. Col polymorph is formed), why did the authors expect the coexistence of the two phases on heating at a rate of 50 °Cmin⁻¹?

We assume this rate is high enough for the formation of some microfilaments and this would cause the coexistence of both phases.

- Assuming that both polymorphs are energetically so close and both lead to liquid at similar temperatures, why do authors expect neat peaks for each potential transition?

Since we do not see any each potential transition, the coexistence of energetically close polymorphs (forming a mixture) appears to be the cause for the broadened phase transitions at lower temperatures.

- Broad peaks and sometimes, lower transition temperatures, can be expected on increasing rates in DSC measurements.

Could be completely ruled out the coexistence of both polymorphs on samples? Could this sort of coexistence lead to broad peak as it is very often observed for mixtures?

We think the referee is correct. The coexistence of both polymorphs, considering sample treatment, appears logical and SAXS data confirm this coexistence for insufficiently rapid cooling rates to lead exclusively to the formation of the B4 microfilament phase.